# Eliciting preferences of persons with dementia and informal caregivers to support ageing in place in the Netherlands: a protocol for a discrete choice experiment

Isabelle Vullings [1,2] Joost Wammes,[1,2] Özgül Uysal-Bozkir,[3] Carolien Smits,[4] Nanon H M Labrie,[5] J D Swait,[6,7] Esther de Bekker-Grob,[6,7] Janet L Macneil-Vroomen[1,2]

For numbered affiliations see end of article.

**Correspondence to**
Isabelle Vullings;
i.vullings@amsterdamumc.nl

## ABSTRACT

**Introduction** Ageing in place (AIP) for persons with dementia is encouraged by European governments and societies. Healthcare packages may need reassessment to account for the preferences of care funders, patients and informal caregivers. By providing insight into people's preferences, discrete choice experiments (DCEs) can help develop consensus between stakeholders. This protocol paper outlines the development of a Dutch national study to cocreate a healthcare package design methodology built on DCEs that is person-centred and helps support informal caregivers and persons with dementia to AIP. A subpopulation analysis of persons with dementia with a migration background is planned due to their high risk for dementia and under-representation in research and care.

**Methods and analysis** The DCE is designed to understand how persons with dementia and informal caregivers choose between different healthcare packages. Qualitative methods are used to identify and prioritise important care components for persons with dementia to AIP. This will provide a list of care components that will be included in the DCE, to quantify the care needs and preferences of persons with dementia and informal caregivers. The DCE will identify individual and joint preferences to AIP. The relative importance of each attribute will be calculated. The DCE data will be analysed with the use of a random parameters logit model.

**Ethics and dissemination** Ethics approval was waived by the Amsterdam University Medical Center (W23_112 #23.137). A study summary will be available on the websites of Alzheimer Nederland, Pharos and Amsterdam Public Health institute. Results are expected to be presented at (inter)national conferences, peer-reviewed papers will be submitted, and a dissemination meeting will be held to bring stakeholders together. The study results will help improve healthcare package design for all stakeholders.

## STRENGTHS AND LIMITATIONS OF THIS STUDY

⇒ This will be the first study to include informal caregivers and persons with dementia to identify ageing in place (AIP) preferences that has been codesigned with health insurers, policy-makers, patient advocacy groups, healthcare professionals, researchers, informal caregivers and persons with dementia.
⇒ Innovative and rigorous economic methods will be employed to evaluate AIP preferences.
⇒ Pooling of persons with dementia with different migration background may lead to better understanding of their needs as they are typically under-represented in research.
⇒ A limitation of this study is the generalisability of results as not all migrant groups living in the Netherlands may be included.
⇒ This study is conducted in the Netherlands; findings might be specific to the Dutch Healthcare setting.

## INTRODUCTION

Nearly every country across Europe has reformed its long-term care policy to emphasise ageing in place (AIP) as a way to control costs associated with population ageing.[1] The Netherlands is the highest spender of long-term care per gross domestic product in the world.[2] Most Dutch long-term care services are paid by the national health insurance, with a low percentage of copayments.[3] To cope with the rising long-term care costs, the Netherlands reformed its long-term care policy in 2015.[4] The reform increased informal care duties and encouraged older adults to AIP.[4 5] A substantial component of the Dutch long-term care costs is attributed to dementia care, as it is the most costly disease encompassing 10.6% of the total Dutch healthcare budget.[6] The dementia population is known for its requirement of complex and costly care.[7] However, it remains unknown how AIP has affected this population. In the years after the reform, preliminary results

indicate an increased need for support with crisis situations such as unplanned hospitalisations, acute nursing home admissions and hospital deaths.[8 9] These crises situations are not only costly, but can also compromise the safety and well-being of persons with dementia AIP, in addition to the informal caregivers expected to carry the load.

Despite the need for evidenced-based policy to ensure proper healthcare services, people with dementia and informal caregivers are frequently excluded from policy creation; therefore, services often do not fit with their needs or are not provided at the right time.[10 11] To support successful AIP, identifying the care preferences of persons with dementia and informal caregivers is crucial. While identifying these preferences, special attention should be given to persons with dementia and informal caregivers with a migration background, who are known to have a higher risk of developing dementia.[12 13] Despite their high risk, persons with a migration background are under-represented in research and care.[14] It is crucial that persons with a migration background be included in research so healthcare package evidence is representative of the dementia population. To provide recommendations for healthcare packages, it is subsequently needed to quantify the identified care preferences, which can be realised with a discrete choice experiment (DCE).

DCEs are a popular stated preference method used to elicit patients' preferences in healthcare on a large scale.[15–17] In a DCE, participants are presented with a series of alternative hypothetical scenarios. Participants are repetitively asked to select in each alternative hypothetical scenario their preferred option from among a presented set of options, for example, care packages.[16] These choice options are characterised by their attributes and corresponding attribute levels.[15 16] An example of an attribute could be emotional support, described as the possibility to talk to someone about personal feelings or concerns. The corresponding attribute levels could be (1) peer support group, (2) psychologist or (3) telephone helpline. The outcomes (choices) from DCEs are analysed based on the assumption that participants act in a utility maximising manner, choosing their most preferred option based on the relative overall attractiveness of the included attributes and attribute levels.[16 18] This protocol paper will outline the development of a Dutch national study to create a healthcare package design methodology, built on DCEs, that is person centred and helps support both informal caregivers and persons with dementia to successfully AIP. Qualitative methods will be used to identify the care needs and preferences of persons with dementia and their informal caregivers. The results of the qualitative studies will provide insights into the attributes (ie, care components) that will be included in the DCE. For the DCE, a subpopulation analysis of persons with dementia with a migration background is planned.

### Aim

The aim of this study is to identify individual and joint preferences of persons with dementia and informal caregivers for in-home support that enables AIP. We also aim to create optimal healthcare packages for persons with dementia and informal caregivers to AIP. For this second aim, choice model inferences will be the basis for creating the most preferred care packages and testing the uptake in a subsample of participants. This will validate the national model inferences and provide policy-makers with high quality, understandable and implementable evidence.

## METHODS
### Overview of the DCE

In this protocol paper, the different phases of designing an inclusive DCE study in which persons with dementia and their informal caregivers can participate will be outlined. The study will be conducted between April 2022 and May 2025. Figure 1 illustrates that the first phase is focused on attribute development, in which semistructured interviews are used to identify important components of care for persons with dementia and informal caregivers AIP. In the second phase, persons with dementia and informal caregivers are asked which of the care components are

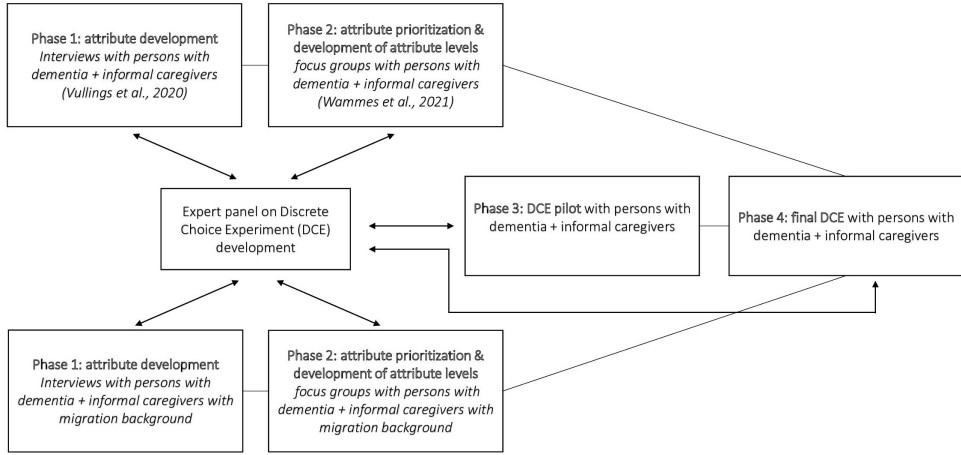

**Figure 1** Flow chart study design to create healthcare packages based on discrete choice experiments.

most important to them, leading to a prioritisation of the attributes and the development of attribute levels. For phases 1 and 2, the Consolidated criteria for Reporting Qualitative research guidelines for reporting qualitative research will be followed to ensure important aspects of the research team, reflexivity, study design, findings and analysis are reported.[19] The third phase consists of a pilot which is used to test if the content of the DCE is understandable for persons with dementia and informal caregivers, and if the list of included attributes and corresponding attribute levels is complete. Finally, in phase 4, the researchers conduct the DCE study. An expert panel will be consulted during all the different phases of the study to reflect on the research design and findings. Phases 3 and 4 will be reported according to the Strengthening the Reporting of Observational Studies in Epidemiology guidelines, developed as a tool for authors to ensure high-quality reporting of observational studies.[20] For constructing the DCE design, the International Society for Pharmacoeconomics and Outcomes Research (ISPOR) good research practices reports will also be followed.[21 22]

## Patient and public involvement

The DCE, including the framing of the questions, will be developed after focus group discussion and multiple individual interviews with persons with dementia and informal caregivers. Persons with dementia, informal caregivers and policy-makers are involved from the start of the study to ensure the appropriateness of the methodology and to help come to meaningful and implementable results.[23] Study materials, such as interview guides, will be discussed with persons with dementia and informal caregivers before the start of the study to ensure appropriateness and make necessary changes. Prior to the DCE, person with dementia, informal caregivers and patient advocacy groups will be asked to voice their opinion about the final list of attributes and corresponding levels included in the DCE. Patient advocacy groups, persons with dementia and informal caregivers, are to be included to make sure that the study properly represents these groups, and the study results reflect their needs and preferences. The patient advocacy groups and policy-makers are consulted through an expert panel, details of which can be found in the next section. The Guidance for Reporting Involvement of Patients and the Public 2 (GRIPP2) checklist will be followed to report the patient and public involvement.[24]

The DCE study will identify individual and joint preferences to AIP, allowing for this concept to be implemented elsewhere. The findings of this study will be published in peer-reviewed journals and an easy-to-understand summary of the results will be available through patient organisations such as Alzheimer Nederland and Pharos. Furthermore, the researchers will organise informative events to communicate the study results to persons with dementia and informal caregivers. These events will be organised together with the patient advocacy groups and

informal caregivers. Finally, a policy brief will be written to ensure that the study results are known to policy creators and can help to improve healthcare package design for older adults and informal caregivers.

## Expert panel

Throughout the entire study, an expert panel will be consulted on the design and content of the DCE, and on the inferences made from its execution. The expert panel will include researchers who have experience with doing DCEs, policy-makers, health insurers, patient advocacy groups, persons with dementia and informal caregivers. The experienced researchers will be asked to provide guidance and feedback on the methodological choices made in the DCE design. Subsequently, the envisioned outcome of the DCE study will be discussed with policy-makers and health insurers to gain insights concerning the comprehension and appropriateness of this envisioned outcome. The inclusion of policy-makers and health insurers in this phase will allow for the opportunity to make changes to the study design that help to realise implementation.

## Participants

This study aims to identify preferences of persons with dementia and informal caregivers AIP; therefore, persons with dementia who live in a long-term care facility will not be included in this study. Additional exclusion criteria will consist of (1) persons who are cognitively impaired to the extent that no conversation can be held with them; (2) people who are unable to provide informed consent. It is know that dementia diagnoses are often delayed, especially in migration populations[25]; therefore, people without a formal diagnosis will not be excluded a priori. However, recruiting participants through healthcare professionals, day-care centres and social organisations that serve persons with dementia and informal caregivers will ensure that all persons have a dementia indication. For the final DCE, the telephone-based interview for cognitive screening (TICS) will be administered to describe the study participants.[26]

Prior to the interviews, focus groups, or DCE choice tasks, the researchers will go through the information letter and informed consent form together with the participants to ensure proper understanding and provide the opportunity to ask for clarifications. Subsequently, participants will be asked to provide written informed consent. If participants are unable to write, verbal continued informed consent will be used.[27] Throughout the study, participants will be continuously reminded that they are participating in research to ensure that they remain informed and are comfortable to continue.[27] It will be stressed that participation can be stopped at any moment.

## Attribute and level development for Dutch persons with dementia and informal caregivers

Study phases 1 and 2 are already partly completed, as interviews and focus groups were held with persons with

**Table 1** Attribute description and corresponding levels

| Attribute | Description | Possible levels |
|---|---|---|
| In-home care | I can get assistance at home with personal care such as showering, dressing, or medication | ▶ Daily on a fixed time<br>▶ 24/7 on demand |
| Help with daily activities | I can get assistance at home with household tasks such as groceries, laundry, cooking, cleaning, or help with doing my finances | ▶ Once per week<br>▶ Multiple times per week |
| Social activities | I can participate in social activities that I like to do | ▶ At a day-care facility<br>▶ At home |
| Emotional support | I can talk to someone when I feel down or want to share my worries | ▶ Peer support group<br>▶ Psychologist<br>▶ Telephone helpline<br>▶ Case manager |
| Information about dementia | I can get information about having dementia | ▶ Telephone helpline<br>▶ Case manager |
| Navigating the healthcare system | I can get assistance with organising care, and help with insurances | ▶ Telephone helpline<br>▶ Case manager |
| Home adaptations and tools | I can get home adaptations and tools such as a stair lift, grips in shower and toilet, or a personal alarm | ▶ No reimbursements<br>▶ Fully reimbursed |

Current list of attributes may need to be updated after expert panel review and interviews with persons with dementia and informal caregivers with a migration background.

dementia and informal caregivers across the Netherlands to identify attributes and levels (table 1).[28 29] The identified attributes and levels were similar for persons with dementia and informal caregivers; however, they prioritised the care components differently.[29] People with dementia prioritised day-to-day help and social care, while for informal caregivers, priorities were information about dementia, organisation of care and emotional support.[29] Unfortunately, no persons with a migration background were included in these qualitative studies. Therefore, the interviews and focus groups will be repeated with persons with dementia and informal caregivers with a migration background to ensure the attributes and levels are applicable to this part of society as they have a higher risk for dementia compared with people with non-migration background.[12]

### Attribute identification for persons with dementia and informal caregivers with a migration background

To identify care preferences and needs of persons with dementia and informal caregivers with a migration background, semistructured interviews will be performed across the Netherlands. Participants will primarily include persons with dementia and informal caregivers with a Turkish, Moroccan or Surinamese background as these are the largest groups of non-western immigrants in the Netherlands and they are known to have a high risk for developing dementia.[12] People living with dementia who do not speak Dutch as their first language will be welcome to participate.

Participants will be recruited with the help of day-care centres, general practitioners and organisations that serve persons with a migration background (Pharos and Netwerk van Organisaties van Ouderen Migranten

(NOOM)). The organisations will inform potential participants about the study, and interested participants will be provided with a bilingual invitation letter. The interviews will be conducted in the preferred language of the participants: Turkish, Moroccan-Arabic, Tarifit or Dutch.[12] Interviews will be conducted by trained bilingual interviewers. Persons with dementia and informal caregivers will be interviewed separately. However, since it is important that participants feel comfortable, if they strongly wish to be interviewed together this will be permitted.

The interviews will be semistructured, which means that an interview guide (online supplemental file 1) will be used to provide some structure, while leaving room for new topics to be brought up by the participants. The interview guide will generally be based on the guide (online supplemental file 2) of the interview study for persons with non-migration backgrounds,[28] with additions based on findings of previous studies that focused on persons with dementia and informal caregivers with a migration background.[30 31] The cultural sensitivity and understandability of the interview guide will be assessed by Pharos, centre of expertise in health disparities and experienced researchers (CS and ÖUB). The interviews will be transcribed and translated by the bilingual interviewer who conducted the interviews. Respondents will have the opportunity to review the translated transcripts, which will help to ensure proper interpretation and provide opportunities to give clarifications. It will be stressed that this is voluntary; participants are free to choose if they wish to read the translated transcripts. Interviews will be held until data saturation is reached, which is expected after approximately 10 interviews for each group.[32] Reflexive thematic analysis will be used to

analyse and identify relevant patterns within the data.[33][34] Coding of the data will be done with both a deductive and inductive approach, starting with certain theoretical or empirical assumptions, with an open approach.[33] The results will provide a list of attributes (care components) that are important for persons with dementia and informal caregivers with a migration background.

## Attribute prioritisation and the development of attribute levels

To understand the relative importance of the care components that are identified, mixed focus group sessions will be held with a new sample of persons with dementia and informal caregivers with a Turkish, Moroccan or Surinamese background.[12] Focus groups create an environment in which participants can share their experiences and build on each other's knowledge.[29] To ensure a free discussion, informal caregivers will not be participating in the same focus group as their person with dementia.[29] Participants will be recruited with the help of previously mentioned organisations. To facilitate dialogue between the participants, focus groups will include participants with similar migration backgrounds and linguistic preferences. The focus groups will be conducted by trained bicultural interviewers. The focus group session will include a quantitative ranking exercise in which persons with dementia and informal caregivers will be asked to list the care components from most important to least important.[29] For this ranking exercise, cards with the previously identified care characteristics, containing visual and written information will be used. During the focus group, the reasoning behind the choices made in the ranking exercise will be discussed.

Focus groups will be held until saturation is reached, which is expected after approximately six sessions, each including persons with dementia (n=4) and informal caregivers (n=4). Focus groups will be transcribed and translated by the trained bilingual interviewers. Participants will be provided with the opportunity to read the translated transcripts and, if necessary, provide clarifications. Reflexive thematic analysis will be used to analyse and identify relevant patterns within the data.[33][34] Coding of the data will be done with both a deductive and inductive approach, starting with certain theoretical or empirical assumptions, with an open approach.[33] The results will consist of a ranking of the attributes (care components) and a description of the attribute levels (characteristics of care components).

## DCE pilot

For the DCE pilot, the list of attributes that has previously been identified (table 1) will be updated with the findings of the qualitative studies with persons with dementia and informal caregivers with a migration background. The pilot will include both individual DCE rounds with the persons with dementia (n=4) and informal caregivers (n=4), and dyadic rounds with both the person with dementia and their informal caregiver. The in-person pilot will include the testing of respondents' understanding

of choice tasks and the appropriateness of the included attributes and attribute levels. Furthermore, the pilot will help to illustrate if the length of the DCE survey is acceptable. Based on the pilot, changes might need to be made to the DCE survey to increase appropriateness and understandability, as well as manage survey complexity for persons with dementia.

## The final DCE

The final DCE will include three rounds: (1) persons with dementia, (2) informal caregivers and (3) dyadic rounds with both the persons with dementia and informal caregivers. The first two rounds will help to elucidate individual preferences, while the dyadic round will help to elucidate joint preferences. Identifying joint preferences is important since decisions about care are often made by multiple individuals.[35–37] Furthermore, the dyadic DCE round can pose as an opportunity for the person with dementia and their informal caregiver to have an in-depth conversation about their preferences and needs, which could lead to better mutual understanding.[38] It is likely that preference adaptation will occur; however, as shown in previous research, this adaptation can be from both the person with dementia and the informal caregiver.[38] Informal caregivers were not primarily dominant in dyadic DCE rounds, in fact the informal caregivers were found to be helpful in assisting the person with dementia to complete more choice tasks.[38] An active role of the researcher administering the DCE choice tasks is needed to help guide the dialogue in the joint DCE rounds. People with dementia will be encouraged to state their preference and motivation first to minimise agency of informal caregivers.[38]

The complexity of a DCE can be challenging for persons with dementia.[39–41] However, a recent study found that persons with moderate cognitive impairment can complete DCE choice tasks.[38] Therefore, persons will not be excluded a priori based on their level of cognitive impairment. The following options in the DCE design will help to lower the cognitive burden for persons with dementia. First, illustrations will be used to visualise the choice sets and included attributes. Figure 2 provides an example of such an illustration.[38] The illustrations are made by a graphic designer who has experience illustrating pictures for persons with cognitive impairments. Second, the DCE will have binary choice tasks, asking participants to choose between packages A and B, instead of including multiple alternatives (eg, additional packages C or D). Participant will complete a maximum of six choice tasks, consisting of three attributes and two corresponding levels.[38] Blocked fractional factorial designs[42] will be created with the use of Ngene. Finally, the DCE will start with a practice round to test decision making skills.[38] Persons with dementia will be asked to think aloud while completing the choice tasks. The think-aloud technique is commonly used in research to make thought processes of participants observable for researchers.[43] It has been found useful in supporting persons with dementia to

**Figure 2** Illustrated discrete choice experiment choice tasks.

complete DCE choice tasks, as it enables researchers to observe difficulties and, if needed, provide guidance or remind persons with dementia about the rules of the choice tasks.[38]

### Power calculation

Envisioning a two-alternative choice task and four replicates per dyad, a minimum sample size n≥576 is recommended for a DCE to reliably achieve 95% CIs on true uptake probabilities of 0.4.[44] Therefore, the goal will be to achieve a sample size of 600 dyads of persons with dementia and informal caregivers, covering all targeted groups. Purposive sampling of minority groups, people living in urban and rural areas, and people with different educational levels will be used. In addition, sampling across the country will help to make the healthcare package as nationally representative as possible.[45]

Participants will be recruited through general practitioners, geriatricians, day-care facilities and organisations that serve persons with dementia and informal caregivers (Alzheimer Nederland, Pharos, NOOM). The professionals will inform potential participants about the study, and provide them with the (bilingual) information letter. With the participants' consent, the researchers will obtain their contact information form the professionals. The researchers will contact the potential participant by telephone to answer any remaining questions and to administer the TICS.[26] The researchers will schedule an appointment at the participants home to complete the DCE choice tasks in person.

### Outcomes

The main outcome of the DCE study will be the relative importance of the attributes (care components). The relative importance of each attribute will be calculated as the difference between the preference weights of the most and least preferred level of that attribute.[46] The relative importance will be scaled so that 10 indicates the most preferred attribute. The difference between individual

and joint DCE round will be analysed with summary statistics and bivariate comparisons.

A contingent valuation approach will be used to identify the participants' willingness to pay (WTP) for the most preferred programmes outside of the DCE.[47 48] While out-of-pockets costs for social care does not currently occur in the Netherlands yet, this may soon change. Therefore, an indication of the WTP is helpful for policy-makers and health insurers. At the end of the DCE, participants indicate their preferred envisioned package over their current service provision, they will be asked if they are willing to pay an additional monthly out-of-pocket fee of €50 for this package. If participants answer this question with yes, additional questions with out-of-pocket fees of €100 and €200 will be asked.[47] If participants answered no to the initial question of €50, similar questions with an amount of €25 and €12,50 will be asked.[47] Finally, participants will be asked to indicate their maximum WTP in an open-ended question.[47] For the analysis, the maximum price expressed in the open-ended question will be used and summarised through the median and IQR.[47]

### Statistical analysis

The DCE data will be analysed with a random parameters logit (RPL) model.[46] For each attribute level, coefficient estimates will be estimated from the RPL model, which can be interpreted as the relative preference weight.[46] Individual and joint models will be created to evaluate preferences of persons with dementia and informal caregivers at the individual and joint DCE rounds.

To analyse if persons with dementia and informal caregivers with a migration background have different preferences, post hoc subgroup analysis will be conducted. Dummy coded variables will be added to the RPL model to identify participants that are part of given subgroups. The attribute levels in the RPL model will be interacted with the dummy variable, and all interaction terms will be added to the original RPL model.[46] The estimated parameters of the interaction terms will illustrate the difference in preferences between the subgroup (persons with a migration background) and the reference group (persons without a migration background).[46] This method can also be used to analyse if there are different preferences based on other subgroups, such as socioeconomic status.

Finally, the attributes with the highest relative importance can be used to build most preferred programmes (healthcare packages). To ensure validity of these results for health insurers and government, the uptake of the most preferred programmes will be tested on a subsample of the target population. Model predictions will be verified by creating the most preferred programmes according to the preferences of persons with dementia and informal caregivers that were found in the DCE. A new sample will be collected of 50–60 participants to test the uptake of the most preferred programmes by asking participants which one they would rather have over their current service provision.[49]

## DISCUSSION

The discrete choice methodology described in this paper enables persons with dementia and informal caregivers to participate in policy formulation and evaluation. It is assumed that this will help to avoid crisis situations and improve the quality of life of persons with dementia and their informal caregivers. Furthermore, the outcome of the final DCE will be a ranking of the most preferred care components represented in a care package. This helps health insurers and the ministry of health to maximise their chances for designing healthcare packages that are appropriate, useful and meaningful.

Furthermore, this study protocol provides an important methodological contribution. It is a clear guide on how to build an inclusive DCE, covering the important methodological choices throughout the required study phases. Moreover, it provides guidelines on how to include under-represented groups in DCEs. The methodological choices described to help include persons with dementia with a migration background, can be used to include other under-represented groups such as people with low (health) literacy.[50 51] Finally, this study provides an example of how to overcome language barriers in research and knowledge dissemination.

### Strengths and limitations of this study

A strength of this study will be that it is the first study to include informal caregivers and persons with dementia to identify AIP preferences that has been codesigned with health insurers, policy-makers, patient advocacy groups, healthcare professionals, researchers, informal caregivers and persons with dementia. Innovative and rigorous economic methods will be employed to evaluate AIP preferences. Another strength is the inclusion of persons with dementia with different migration backgrounds, which can lead to a better understanding of their needs since they are typically underrepresented in research.

A limitation of this study is the generalisability of results, as not all migrant groups living in the Netherlands may be included. In addition, the chosen recruitment strategies make it unlikely for persons who do not have a diagnoses to be included in this study. Persons with communication difficulties, such as hearing or vision loss, will have more difficulties to participate in DCEs. However, the participants ability to complete the DCE will be assessed with practice choice tasks. Finally, enabling persons to participate from their own homes helps to include persons with limited mobility. The research team strives to make the study as inclusive as possible.

### Ethics and dissemination

Based on the study protocol, the Ethics Committee (METC) of the Amsterdam University Medical Centre waived the obligation for the study to undergo formal ethical approval as is described under Dutch law in the Medical Research in Humans Act, January 2019 (ref W23_112 #23.137).

### Consent

This is a prospective study and pseudonymised data are used; written and continued informed consent will be obtained from the participants prior to participation. This is consistent with current European legislation under the General Data Protection Regulation. This study will abide by the Declaration of Helsinki and present ethical requirements.

**Author affiliations**
¹Geriatrics, Amsterdam UMC Locatie AMC, Amsterdam, Noord-Holland, The Netherlands
²Amsterdam Public Health Research Institute, Amsterdam, North Holland, The Netherlands
³Department of Psychology, Education and Child Studies, Erasmus University Rotterdam, Erasmus School of Social and Behavioural Sciences, Rotterdam, Zuid-Holland, The Netherlands
⁴Pharos Center of Expertise on Health Disparities, Utrecht, The Netherlands
⁵Department of Language, Literature and Communication, Vrije Universiteit Amsterdam, Amsterdam, Noord-Holland, The Netherlands
⁶Erasmus Choice Modelling Centre, Erasmus University Rotterdam, Rotterdam, The Netherlands
⁷Erasmus School of Health Policy & Management, Erasmus University Rotterdam, Rotterdam, The Netherlands

**Contributors** JLM-V conceptualised the study and developed the study design together with the other researchers. IV and JW contributed to the development of the study design, data collection and analysis. The authors provided input for the study design from their own area of expertise; inclusive research (CS and ÖU-B), qualitative research (NHML), discrete choice experiments (EdB-G and JDS). IV wrote the manuscript, the coauthors read, edited and approved the final version.

**Funding** This work was supported by Alzheimer Nederland, (WE,06-2021-04). In addition, grant support was from The Netherlands Organisation for Scientific Research (https://www.nwo.nl/en) (NWO-Talent- Scheme-Vidi-Grant No. 09150171910002) to EdB-G.

**Competing interests** None declared.

**Patient and public involvement** Patients and/or the public were involved in the design, or conduct, or reporting, or dissemination plans of this research. Refer to the Methods section for further details.

**Patient consent for publication** Not applicable.

**Provenance and peer review** Not commissioned; externally peer reviewed.

**ORCID iD**
Isabelle Vullings http://orcid.org/0000-0002-9604-3333

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
