## [Reviewer comments · BMJ Open]

ARTICLE DETAILS

TITLE (PROVISIONAL)	Eliciting preferences of persons with dementia and informal caregivers to support ageing in place in the Netherlands: A protocol for a discrete choice experiment.
AUTHORS	Vullings, Isabelle; Wammes, Joost; Uysal-Bozkir, Özgül; Smits, Carolien; Labrie, Nanon; Swait, J.D.; de Bekker-Grob, Esther; Macneil-Vroomen, Janet

VERSION 1 – REVIEW

REVIEWER	Haroon, Muhammad The University of Queensland, Faculty of Medicine
REVIEW RETURNED	16-Jul-2023

GENERAL COMMENTS	This is a very important line of research. The inclusion of preferences of people living with dementia (especially from culturally and linguistically diverse background) in healthcare policies has always been a challenge and is very much needed. The design of this study is appropriate to achieve this goal. The protocol is well written and this reviewer found it easy-to-read and convincing. No major flaw was found to the best of the knowledge of this reviewer. Some minor clarifications/amendments are suggested below. 1. Page 3 of 15 (Line 8): Suggests inserting the word “support” in the sentence?. The new re-structured sentence will read as: “In the years after the reform, preliminary results indicate an increased need for support with crisis help, unplanned hospitalizations, acute nursing home admissions, and hospital deaths”. Or the word “crisis help” can be replaced with “crisis situations” (since it is used in the subsequent sentence as well) and the re-structured sentence will become: The new re-structured sentence will read as: “In the years after the reform, preliminary results indicate an increased need for support with crisis situations such as unplanned hospitalizations, acute nursing home admissions, and hospital deaths”. I guess that is what the authors want to say.2. Page 3 of 15 (Line 9): The term “raise question” is a bit vague. Can be replaced with the word “compromise”. Perhaps the authors want to say “These crises situations are not only costly, but can also compromise the safety and well-being of persons with dementia ageing in place, in addition to the informal caregivers expected to carry the load”.3. Page 3 of 15 (Line 25-26): A reader with no specialist background might want to know why Discrete Choice Experiment (DCE) ? Are there other choice experiments as well? Why choose
--

	DCE and not the others ? A brief mention of other alternatives to DCE and then a valid reason for selecting DCE will work. 4. Page 4 of 15 (Line 10-11): Suggests an additional sentence explaining why COREQ is important. Just mention the three domains of COREQ (Research Team and Reflexivity, Study Design, and Analysis and Findings). 5. Page 4 of 15 (Line 15-16): Yes there is a mention of expert panel in the 'Note' of the Figure 1, however it does not show that "expert panel will be consulted during ALL the different phases of the study". The mention of figure 1 in these lines is perhaps unnecessary. 6. Page 4 of 15 (Line 18): An additional sentence is needed explaining STROBE a little bit. Just one sentence. 7. Page 4 of 15 (Line 26-31): The inclusion of People living with dementia and their informal care-givers through interviews and focus groups is evident from this paragraph. The involvement of expert panel is also very clear in the paragraph. However, the authors mention that policymakers and patient advocacy groups will also be involved. Since they will be part of the expert panel (and that is discussed in the subsequent section titled 'Expert Panel'). The reviewer suggests to either point towards the next section by including the words "see the next section" or explicitly state that policy-makers and advocacy groups will be part of the expert panel and the details of which can be seen in the subsequent section. 8. Page 5 of 15 (Line 36 – 56): Will there be any additional inclusion/exclusion criteria for the eligibility of the participants particularly for people living with dementia? Some people living with dementia might not be able to communicate especially in the advanced stages. Also, will other age-related difficulties such as hearing and vision loss affect the eligibility of the participants? 9. Page 7 of 15 (Line 25-26): Yes, the assistance of informal caregivers is very important, however that also means that the choices and preferences of people living with dementia are also influenced by their caregivers. How would you address that? Perhaps that can be a limitation of the study. 10. Page 9 of 15 (Line 26-30): Obtaining informed consent from some people living with dementia can be tricky, particularly from people in the advanced stages. Do you have any threshold/exclusion criteria based on communication abilities or stage of dementia? This is again related to the aforementioned inclusion/exclusion criteria (see comment 8). 11. A general comment (yet, important): What are the dates/time frame for the planned studies?
--	--

REVIEWER	WANG , Kailu The Chinese University of Hong Kong
REVIEW RETURNED	21-Jul-2023

GENERAL COMMENTS	This is a research protocol for a discrete choice experiment to elicit preference for ageing in place among persons with dementia (PwD) and informal caregivers. The DCE attributes and levels
--

	were identified from interviews and focus groups of the participants, which may be updated based on feedbacks in expert panels and qualitative studies in participants with migration backgrounds. The protocol presents a good quality of study design. The topics of this study is important in formulating relative packages to enable ageing in place. Nevertheless, there is information to be clarified in methods, particularly in study participants.  1. It is not clear that what settings the participants come from. Are all of them community-dwelling, or part of them have already been admitted to a residential care home? And how they will be approached for DCE in this study? 2. PwD would have different level of cognitive impairment. Which group of PwD in terms of their cognitive impairment would the authors plan to recruit? And how would the authors ascertain the individual is with dementia (doctor-diagnosed or assessed with other instrument)? 3. If the level of cognitive impairment is too severe, PwD may have difficulties to weigh all the alternatives with all the attributes presented to them, even if there are pictures to assist them. Therefore, it is suggested that the authors to define the cognitive impairment level of their prospective participants. 4. It is interesting that there will be a dyadic DCE round to find out their joint preferences. Does the authors have any plan to compare the results between any two of the three rounds of DCE? 5. The statistical analysis for the contingent valuation for WTP should be added in corresponding section as well.
--	--

REVIEWER	O'Shea, Eamon School of Business & Economics, National University of Ireland, Economics
REVIEW RETURNED	02-Aug-2023

GENERAL COMMENTS	This is an interesting and welcome study. The protocol provides a useful, accurate and appropriate account of the work to be undertaken; methods and analysis are well described. The points below are suggestions and questions that the authors may want to consider in any revision and when writing-up the final study. Comments to the authors  1. The work and the survey is focused on aging in place (AIP) for people currently living at home. What about the preferences of people living in places other than their own homes, for example those living in sheltered housing, housing with care and long-stay residential care. More specifically, people on the boundary of care at different transition points on the continuum of care may have stronger preferences than those who have not yet been confronted with moving between locations. 2. The protocol has little to say about the cognitive cut-off point for participation in the study. It is stated that the in-person pilot will include the testing of respondents' understanding of choice tasks, but given the heterogeneity of the dementia experience is that enough to ensure comprehensive coverage for the survey?
--

	3. Why was ethical approval waived by the Amsterdam University Medical centre? 4. I would like to see more granular and detailed description of the PPI work over the course of the study, through to dissemination and policy dialogue work. How inclusive were the original PPI deliberations? What was the reach of the PPI participants in terms of the severity of their cognitive impairment, income levels, family circumstances, location, gender? What about potential biases of those proposing PPI participants? 5. The authors say that survey participants will be recruited with the help of organisations that serve person with dementia and their informal caregivers, but we need to hear more about the potential bias associated with that recruitment method. What about those people with dementia that remain undiagnosed and therefore lie beyond the ambit and knowledge of local and national organizations. Moreover, some people with dementia are hard to reach even with a diagnosis, particularly those with severe communication difficulties and those with aggressive behaviours. 6. In relation to Table 1 on attributes, I was not always sure to whom the attributes referred - person with dementia or carers. Specifically, are the emotional support, organisation of care and information about dementia attributes prioritised exclusively/mainly/ by carers or were they also referenced by people with dementia. Some more clarity is need on this Table. 7. The final list of attributes and levels will be even more complicated and complex following input from/about persons with dementia and informal caregivers with a migration background. I would have concerns about imposing an even greater cognitive load on the respondents, notwithstanding potential steps take by the authors to alleviate it through illustrations and other mechanisms. 8. The dyadic DCE round will help to explore joint decision-making and the comparisons to singular decision-making will be interesting, but these conversations also require great skill and needs to be described more in the methods. The dominance of the carer still needs to be monitored, irrespective of recent evidence; so too does the quality of the relationship between carer and person with dementia. Moreover, making sure that the carer in the conversation is indeed the correct (or only) joint-decision-maker is also relevant for preference revelation. 9. Are respondents in Holland currently asked to pay out-of-pocket for social care provision. If so, the WTP questions make sense, but become problematic if people are not used to valuing their own care in that monetised way once insurance contributions have been paid? 10. I think resource allocation decision-making for aging in place in Holland needs more elaboration in the protocol – reader needs to know a little more on provision and financing. It is an interesting question as to whose preferences matter most in such circumstances – taxpayers/ social insurance contributors or care recipients, given the implications for the former in relation to tax/social insurance contributions. Designing health care packages that are appropriate, useful and meaningful costs money and therefore the general public may need to be consulted on the financial implications of expanding/enhancing provision.
--	--

VERSION 1 – AUTHOR RESPONSE

Reviewer: 1

Dr. Muhammad Haroon, The University of Queensland

Comments to the Author:

This is a very important line of research. The inclusion of preferences of people living with dementia (especially from culturally and linguistically diverse background) in healthcare policies has always been a challenge and is very much needed. The design of this study is appropriate to achieve this goal. The protocol is well written and this reviewer found it easy-to-read and convincing. No major flaw was found to the best of the knowledge of this reviewer. Some minor clarifications/amendments are suggested below.

Dear Dr. Muhammad Haroon, thank you for your time reviewing our protocol paper, and for all the helpful comments and suggestions.

Comment 1

Page 3 of 15 (Line 8): Suggests inserting the word “support” in the sentence?. The new re-structured sentence will read as: “In the years after the reform, preliminary results indicate an increased need for support with crisis help, unplanned hospitalizations, acute nursing home admissions, and hospital deaths”. Or the word “crisis help” can be replaced with “crisis situations” (since it is used in the subsequent sentence as well) and the re-structured sentence will become: The new re-structured sentence will read as: “In the years after the reform, preliminary results indicate an increased need for support with crisis situations such as unplanned hospitalizations, acute nursing home admissions, and hospital deaths”. I guess that is what the authors want to say.

Answer 1

Thank you for this comment. We revised this sentence based on your second suggestion.

See introduction, page 3

“In the years after the reform, preliminary results indicate an increased need for support with crisis situations such as unplanned hospitalizations, acute nursing home admissions, and hospital deaths⁸9”.

Comment 2

Page 3 of 15 (Line 9): The term “raise question” is a bit vague. Can be replaced with the word “compromise”. Perhaps the authors want to say “These crises situations are not only costly, but can also compromise the safety and well-being of persons with dementia ageing in place, in addition to the informal caregivers expected to carry the load”.

Answer 2

We replaced “raise question” with “compromise”

See introduction, page 3

“These crises situations are not only costly, but can also compromise the safety and well-being of persons with dementia ageing in place, in addition to the informal caregivers expected to carry the load”.

Comment 3

Page 3 of 15 (Line 25-26): A reader with no specialist background might want to know why Discrete Choice Experiment (DCE) ? Are there other choice experiments as well? Why choose DCE and not the others ? A brief mention of other alternatives to DCE and then a valid reason for selecting DCE will work.

Answer 3

We have added a citation for the reader to get a better foundation on our rationale and revised the sentence to explain DCEs are a form of stated preference methods. DCEs are of the most popular methods within econometrics to study the distribution of choices within a population.

See introduction, page 3

“DCEs are a popular stated preference method used to elicit patients’ preferences in healthcare on a large scale¹⁵⁻¹⁷.”

Comment 4

Page 4 of 15 (Line 10-11): Suggests an additional sentence explaining why COREQ is important. Just mention the three domains of COREQ (Research Team and Reflexivity, Study Design, and Analysis and Findings).

Answer 4

We agree that it is helpful to provide details on the COREQ guidelines, and included this in the methods.

See methods, page 4

“For phases one and two, the COREQ guidelines for reporting qualitative research will be followed to ensure important aspects of the research team, reflexivity, study design, findings, and analysis are reported¹⁹”

Comment 5

Page 4 of 15 (Line 15-16): Yes there is a mention of expert panel in the ‘Note’ of the Figure 1, however it does not show that “expert panel will be consulted during ALL the different phases of the study”. The mention of figure 1 in these lines is perhaps unnecessary.

Answer 5

The expert panel is visualized in Figure 1 in the middle box ‘expert panel on Discrete Choice Experiment development’. The arrows to the different study phases of designing the DCE are there to illustrate that the expert panel is consulted for all these different phases. We hope that the Figure is clear. We agree that it is not necessary to mention Figure 1 twice in this paragraph, and only keep the initial reference to the Figure.

See methods, page 4

“An expert panel will be consulted during all the different phases of the study to reflect upon the research design and findings.”

Comment 6

Page 4 of 15 (Line 18): An additional sentence is needed explaining STROBE a little bit. Just one sentence.

Answer 6

We agree that it would be useful to elaborate on the STROBE guidelines.

See methods, page 4

“Phases three and four will be reported according to the STROBE guidelines, developed as a tool for authors to ensure high-quality reporting of observational studies²⁰”

Comment 7

Page 4 of 15 (Line 26-31): The inclusion of People living with dementia and their informal care-givers through interviews and focus groups is evident from this paragraph. The involvement of expert panel is also very clear in the paragraph. However, the authors mention that policymakers and patient advocacy groups will also be involved. Since they will be part of the expert panel (and that is discussed in the subsequent section titled ‘Expert Panel’). The reviewer suggests to either point towards the next section by including the words “see the next section” or explicitly state that policy-makers and advocacy groups will be part of the expert panel and the details of which can be seen in the subsequent section.

Answer 7

Thank you for this suggestion. We agree that it is needed to provide some clarifications on how the policy makers and patient advocacy groups are included. We mention that they participate in the expert panel.

See methods, page 4

“The patient advocacy groups and policy makers are consulted through an expert panel, details of which can be found in the next section.”

Comment 8

Page 5 of 15 (Line 36 – 56): Will there be any additional inclusion/exclusion criteria for the eligibility of the participants particularly for people living with dementia? Some people living with dementia might not be able to communicate especially in the advanced stages. Also, will other age-related difficulties such as hearing and vision loss affect the eligibility of the participants?

Answer 8

This research includes people with dementia from all dementia stages. However, persons with dementia must be able to engage in a conversation. We added a paragraph in which we provide more details on our inclusion and exclusion criteria. People with hearing loss or visual impairments will not be excluded a-priori, but it will be assessed if they can complete DCEs while performing the practice task.

See methods, page 5

“This study aims to identify preferences of persons with dementia and informal caregivers aging in place, therefore persons with dementia who live in a long-term care facility will not be included in this study. Additional Exclusion criteria will consist of (a) persons who are cognitively impaired to the extent that no conversation can be held with them; (b) people who are unable to provide informed consent. It is known that dementia diagnoses are often delayed, especially in migration populations²⁵, therefore people without a formal diagnoses will not be excluded a-priori. However, recruiting participants through healthcare professionals, day-care centers and social organizations that serve persons with dementia and informal caregivers will ensure that all persons have a dementia indication.”

See discussion, page 10

“Persons with communication difficulties, such as hearing or vision loss will have more difficulties to participate in DCEs. However, the participants ability to complete the DCE will be assessed with practice choice tasks. Finally, enabling persons to participate from their own homes helps to include persons with limited mobility. The research team strives to make the study as inclusive as possible.”

Comment 9.

Page 7 of 15 (Line 25-26): Yes, the assistance of informal caregivers is very important, however that also means that the choices and preferences of people living with dementia are also influenced by their caregivers. How would you address that? Perhaps that can be a limitation of the study.

Answer 9

The DCE study consist of three rounds: (1) person with dementia individually, (2) informal caregiver individually, and (3) person with dementia and informal caregiver jointly. In the individual rounds, people can state their individual preferences. In the joint round, people with dementia and informal caregivers will be encouraged to learn and discuss each other's preferences. It is likely preference adaptation will occur, however as shown in our previous research¹. However, this adaptation can come from both the person with dementia and the informal caregiver. During the joint round discussion, people with dementia will be encouraged to state their preference and motivation first to minimize agency of informal caregivers.

See methods, page 8

"It is likely that preference adaptation will occur, however as shown in previous research this adaptation can be from both the person with dementia and the informal caregiver³⁷. Informal caregivers were not primarily dominant in dyadic DCE rounds, in fact the informal caregivers were found to be helpful in assisting the person with dementia to complete more choice tasks³⁷. An active role of the researcher administering the DCE choice tasks is needed to facilitate the dialogue in the joint DCE rounds. People with dementia will be encouraged to state their preference and motivation first to minimize agency of informal caregivers³⁷.

Comment 10.

Page 9 of 15(Line 26-30): Obtaining informed consent from some people living with dementia can be tricky, particularly from people in the advanced stages. Do you have any threshold/exclusion criteria based on communication abilities or stage of dementia? This is again related to the aforementioned inclusion/exclusion criteria (see comment 8).

Answer 10

Thank you for this comment. We agree that obtaining informed consent from persons with dementia can be difficult and that further explanation is needed. We will read the information letter and informed consent form together with the participants, to make sure that they properly understand it and have the opportunity to ask questions. Afterwards, we ask participants to provide written informed consent. In case participants are unable to write, we will record a verbal informed consent. In addition, continuous consent will be used throughout the study to make sure that participants remain informed. Participants who are unable to provide informed consent will be excluded from the study, we added this to our exclusion criteria.

See methods, page 5

"Additional Exclusion criteria will consist of (a) persons who are cognitively impaired to the extent that no conversation can be held with them; (b) people who are unable to provide informed consent.

See methods, page 5

"Prior to the interviews, focus groups, or DCE choice tasks, the researchers will go through the information letter and informed consent form together with the participants to ensure a proper understanding and provide the opportunity to ask for clarifications. Subsequently, participants will be asked to provide written informed consent. If participants are unable to write, verbal continued informed consent will be used²⁷. Throughout the study, participants will be continuously reminded

that they are participating in research to ensure that they remain informed and are comfortable to continue²⁷. It will be stressed that participation can be stopped at any moment.”

Comment 11.

A general comment (yet, important): What are the dates/time frame for the planned studies?

Answer 11

Time frame is added in the methods section.

See methods, page 4

“The study will be conducted between April 2022 and May 2025.”

Reviewer: 2

Dr. Kailu WANG , The Chinese University of Hong Kong

Comments to the Author:

This is a research protocol for a discrete choice experiment to elicit preference for ageing in place among persons with dementia (PwD) and informal caregivers. The DCE attributes and levels were identified from interviews and focus groups of the participants, which may be updated based on feedbacks in expert panels and qualitative studies in participants with migration backgrounds. The protocol presents a good quality of study design. The topics of this study is important in formulating relative packages to enable ageing in place. Nevertheless, there is information to be clarified in methods, particularly in study participants.

Dear Dr. Kailu Wang, thank you for your time reviewing our protocol paper, and for all the helpful comments and suggestions.

Comment 1

It is not clear that what settings the participants come from. Are all of them community-dwelling, or part of them have already been admitted to a residential care home? And how they will be approached for DCE in this study?

Answer 1

Thank you for this comment. We wish to gain insight in care needs and preferences of these persons with dementia who are living at home (or in other words aging in place) and their informal caregivers. We added a paragraph about the study participants in which we provide the in- and exclusion criteria.

We will contact informal caregivers and persons with dementia through healthcare- professionals and organizations, social organizations, and groups. We added details about recruitment in the methods.

See methods, page 5

“The study aims to identify preferences of persons with dementia and informal caregivers aging in place, therefore persons with dementia who live in a long-term care facility will not be included in this study.”

See methods, page 8-9

“Participants will be recruited through general practitioners, geriatricians, day-care facilities, and organizations that serve persons with dementia and informal caregivers (Alzheimer Nederland, Pharos, NOOM). The professionals will inform potential participants about the study, and provide them with the (bilingual) information letter. With the participants’ consent, the researchers will obtain

their contact information from the professionals. The researchers will contact the potential participant by telephone to answer any remaining questions and to administer the TICS26. The researchers will schedule an appointment at the participants home to complete the DCE choice tasks in person.”

Comment 2

PwD would have different level of cognitive impairment. Which group of PwD in terms of their cognitive impairment would the authors plan to recruit? And how would the authors ascertain the individual is with dementia (doctor-diagnosed or assessed with other instrument)?

Answer 2

See reviewer 1 comment 8

Comment 3

If the level of cognitive impairment is too severe, PwD may have difficulties to weigh all the alternatives with all the attributes presented to them, even if there are pictures to assist them. Therefore, it is suggested that the authors to define the cognitive impairment level of their prospective participants.

Answer 3

In a previous mixed method study by our colleague J. Wammes¹ it was found that there is indeed a positive association between cognitive screening test scores and the maximum number of choice tasks completed and the maximum number of attributes within the choice tasks. However, it was found that even persons with moderate cognitive impairment could complete simple choice tasks. Therefore, we do not want to exclude people based on the level of cognitive impairment. We do administer the Cognitive Screening test (Telephone Interview Cognitive Status) to have a description of our study population.

In addition, we make several choices in the DCE design that help to lower the cognitive burden for persons with dementia.

See methods, page 5

“For the final DCE, the telephone-based interview for cognitive screening (TICS) will be administered to describe the study participants²⁶”

See methods, page 8

“The complexity of a DCE can be challenging for persons with dementia³⁸⁻⁴⁰. However, a recent study found that persons with moderate cognitive impairment can be able to complete DCE choice tasks³⁷. Therefore, persons will not be excluded a-priori based on their level of cognitive impairment. The following options in the DCE design will help to lower the cognitive burden for persons with dementia. First, illustrations will be used to visualize the choice sets and included attributes. Figure 2 provides an example of such an illustration³⁷. The illustrations are made by a graphic designer who has experience illustrating pictures for persons with cognitive impairments. Second, the DCE will have binary choice tasks, asking participants to choose between packages A and B, instead of including multiple alternatives (e.g., additional packages C or D). Participant will complete a maximum of six choice tasks, consisting of three attributes and two corresponding levels³⁷. Blocked fractional factorial designs⁴¹ will be created with the use of Ngene. Finally, the DCE will start with a practice round to test decision making skills and persons with dementia will be asked to think aloud whilst completing the choice tasks. The think-aloud technique is commonly used in research to make thought processes of the participants observable for the researchers⁴¹. It has been found particularly useful in supporting persons with dementia to complete DCE choice tasks, as it enables researchers to observe difficulties and, if necessary, provide extra guidance or remind persons with dementia about the rules of the choice tasks³⁷.”

Comment 4

It is interesting that there will be a dyadic DCE round to find out their joint preferences. Does the authors have any plan to compare the results between any two of the three rounds of DCE?

Answer 4

We will compare individual rounds and the joint round with the use of summary statistics and bivariate comparisons.

See methods, page 9

“The difference between individual and joint DCE rounds will be analysed with summary statistics and bivariate comparisons.”

Comment 5

The statistical analysis for the contingent valuation for WTP should be added in corresponding section as well.

Answer 5

For the analysis of the WTP, the maximum prices expressed in the final open-ended question of the contingent valuation approach will be summarized with the median and interquartile range.

See methods, page 9

“For the analysis, the maximum price expressed in the open-ended question will be used and summarized through the median and interquartile range⁴⁶.”

Reviewer: 3

Dr. Eamon O'Shea, School of Business & Economics, National University of Ireland

Comments to the Author:

See attached file

This is an interesting and welcome study. The protocol provides a useful, accurate and appropriate account of the work to be undertaken; methods and analysis are well described. The points below are suggestions and questions that the authors may want to consider in any revision and when writing-up the final study. Comments to the authors

Dear Dr. Eamon O'Shea, thank you for your time reviewing our protocol paper, and for all the helpful comments and suggestions.

Comment 1.

The work and the survey is focused on aging in place (AIP) for people currently living at home. What about the preferences of people living in places other than their own homes, for example those living in sheltered housing, housing with care and long-stay residential care. More specifically, people on the boundary of care at different transition points on the continuum of care may have stronger preferences than those who have not yet been confronted with moving between locations.

Answer 1

Thank you for this comment. We chose to focus specially on persons aging in place, since it was the aim of the 2015 Dutch long-term care reform that older adult stay at home as long as possible. Since the reform only people that need 24/7 care or supervision can be admitted to sheltered housing or

long-term care facilities. Therefore, we focus on people who are still living at home. We do include people who are thinking about moving to a long-term care facility, or are on the waiting list. Identifying care preferences of person who are living in the nursing home is beyond the scope of our current study, as needs and preferences of this population are expected to be different from those who reside at home.

Comment 2.

The protocol has little to say about the cognitive cut-off point for participation in the study. It is stated that the in-person pilot will include the testing of respondents' understanding of choice tasks, but given the heterogeneity of the dementia experience is that enough to ensure comprehensive coverage for the survey?

Answer 2

Due to the heterogeneity in cognitive and DCE decision-making abilities in persons with dementia¹ we do not a-priori exclude people on cognitive screening test score, but merely use these tests to provide a description of the study population. Through recruitment strategies we try to include people that can engage in conversations and complete DCE tasks. Practice DCE tasks will be performed to check DCE choice decision-making abilities, to ensure robust choice task outcomes.

See reviewer 2, comment 3

Comment 3.

Why was ethical approval waived by the Amsterdam University Medical centre?

Answer 3

A waiver was obtained by the Ethics committee of the Amsterdam UMC, what stated that the study does not fall under the Medical Research involving Human Subjects Act, which means no further ethical approval is not necessary. The Medical Research involving Human Subjects Act (WMO) applies if 1) it concerns medical scientific research and 2) participants are subject to procedures or are required to follow rules of behaviors.

Comment 4

I would like to see more granular and detailed description of the PPI work over the course of the study, through to dissemination and policy dialogue work. How inclusive were the original PPI deliberations? What was the reach of the PPI participants in terms of the severity of their cognitive impairment, income levels, family circumstances, location, gender? What about potential biases of those proposing PPI participants?

Answer 4

Thank you for this comment. We included a more detailed description of the PPI work over the course of the study. We agree that it is important to consider the inclusiveness of the PPI work, for this reason we have worked together with organizations that focus on health inequalities and inclusive research such as Pharos and NOOM. These organizations will help with the recruitment of participants for our study and the PPI work, which hopefully helps to enable a more inclusive panel.

See methods, page 4

“Study materials, such as interview guides, will be discussed with persons with dementia and informal caregivers before the start of the study to ensure appropriateness and make necessary changes. Prior to the DCE, person with dementia, informal caregivers, and patient advocacy groups will be asked to voice their opinion about the final list of attributes and corresponding levels included in the DCE.”

See methods, page 4

“Furthermore, the researchers will organize informative events to communicate the study results to persons with dementia and informal caregivers. These events will be organized together with the patient advocacy groups and informal caregivers.”

Comment 5

The authors say that survey participants will be recruited with the help of organizations that serve person with dementia and their informal caregivers, but we need to hear more about the potential bias associated with that recruitment method. What about those people with dementia that remain undiagnosed and therefore lie beyond the ambit and knowledge of local and national organizations. Moreover, some people with dementia are hard to reach even with a diagnosis, particularly those with severe communication difficulties and those with aggressive behaviors.

Answer 5

Including people with dementia without a diagnoses is difficult due to the recruitment strategy. We have listed this as a limitation of our study. Regarding those with severe communication difficulties and those with aggressive behaviors please see comment 8 reviewer 1.

See discussion, page 10

“In addition, the chosen recruitment strategies make it unlikely for persons who do not have a diagnoses to be included in this study.”

Comment 6

In relation to Table 1 on attributes, I was not always sure to whom the attributes referred - person with dementia or carers. Specifically, are the emotional support, organisation of care and information about dementia attributes prioritised exclusively/mainly/ by carers or were they also referenced by people with dementia. Some more clarity is need on this Table.

Answer 6

The attributes can refer to both the person with dementia and informal caregiver. The attributes organizing care, information, and emotional support were found to be more important for informal caregivers in a prior mixed method study, however they were also discussed by the persons with dementia². For example, persons with dementia often referend to the fact that other people helped them with organizing care, often being the informal caregiver.

See methods, page 5

“The identified attributes and levels were similar for persons with dementia and their informal caregivers, however they prioritized the care components differently²⁹.”

Comment 7

The final list of attributes and levels will be even more complicated and complex following input from/about persons with dementia and informal caregivers with a migration background. I would have concerns about imposing an even greater cognitive load on the respondents, notwithstanding potential steps taken by the authors to alleviate it through illustrations and other mechanisms.

Answer 7

We provide a more elaborate description on how we will minimize the cognitive load for persons with dementia. We aim to give participants a maximum number of six choice tasks, consisting of a maximum of three attributes, with maximum two corresponding attribute levels, which was found to be

a best practice in our prior study¹. This will be done by “attribute blocking”, which refers to only showing a subset of the attributes. The fractional factorial design will be created with the use of Ngene.

See methods, page 8

“Participant will complete a maximum of six choice tasks, consisting of three attributes and two corresponding levels³⁷. Blocked fractional factorial designs⁴¹ will be created with the use of Ngene.”

Comment 8

The dyadic DCE round will help to explore joint decision-making and the comparisons to singular decision-making will be interesting, but these conversations also require great skill and needs to be described more in the methods. The dominance of the carer still needs to be monitored, irrespective of recent evidence; so too does the quality of the relationship between carer and person with dementia. Moreover, making sure that the carer in the conversation is indeed the correct (or only) joint-decision-maker is also relevant for preference revelation.

Answer 8

In the joint round, people with dementia and informal caregivers will be encouraged to learn and discuss each other’s preferences. It is likely preference adaptation will occur, however as shown in our previous research¹ this adaptation can come from both the person with dementia and the informal caregiver. During the joint round discussion, people with dementia will be encouraged to state their preference and motivation first to minimize agency of informal caregivers

Furthermore, it is true that there might be other or more informal caregivers who are relevant in the situation of the person with dementia. For now, we focus on the primary informal caregiver, however; we understand that persons with dementia often have a more complex network of informal caregivers who are involved.

See methods, page 8

“It is likely that preference adaptation will occur, however as shown in previous research this adaptation can be from both the person with dementia and the informal caregiver³⁷. Informal caregivers were not primarily dominant in dyadic DCE rounds, in fact the informal caregivers were found to be helpful in assisting the person with dementia to complete more choice tasks³⁷. An active role of the researcher administering the DCE choice tasks is needed to help guide the dialogue in the joint DCE rounds. People with dementia will be encouraged to state their preference and motivation first to minimize agency of informal caregivers³⁷.”

Comment 9.

Are respondents in Holland currently asked to pay out-of-pocket for social care provision. If so, the WTP questions make sense, but become problematic if people are not used to valuing their own care in that monetised way once insurance contributions have been paid?

Answer 9

In the Netherlands most long-term care services are paid by the national health insurance. For in-home care services there is a fixed monthly payment of 19,- euro regardless of type of service or hours of service used. Only a small proportion of the populations uses a personal budget (monthly fixed budget from the national health insurance) to purchase care. For our study, taking the research questions and current health policy into consideration, we decided that the cost attribute does not belong in the DCE. That said, although out-of-pockets costs for social care is not realistic in the Netherlands yet, this might become realistic soon. Therefore, we would like to have the contingent valuation in the protocol (We agree to have it rather at the end than at the beginning of the

questionnaire to avoid CV influencing DCE outcomes). Noteworthy, irrespective of DCE or CV, we are always cautious in interpreting the WTP results as absolute values (in contrast to relative values) for health policy making, if there is no think-aloud, debriefing or other form of qualitative research or external validity checks.

See methods, page 9

“A contingent valuation approach will be used to identify the participants’ willingness to pay (WTP) for the most preferred programs outside of the DCE46 47. While out-of-pockets costs for social care does not currently occur in the Netherlands yet, this may soon change. Therefore, an indication of the WTP is helpful for policy makers and health insurers.”

Comment 10

I think resource allocation decision-making for aging in place in Holland needs more elaboration in the protocol – reader needs to know a little more on provision and financing. It is an interesting question as to whose preferences matter most in such circumstances – taxpayers/ social insurance contributors or care recipients, given the implications for the former in relation to tax/social insurance contributions. Designing health care packages that are appropriate, useful and meaningful costs money and therefore the general public may need to be consulted on the financial implications of expanding/enhancing provision.

Answer 10

We agree that it is important to provide our readers with more background about service provision and financing in the Netherlands. We included this in the introduction.

See introduction, page 3

“The Netherlands is the highest spender of long-term care per gross domestic product in the world². Most Dutch long-term care services are paid by the national health insurance, with a low percentage of co-payments³. To cope with the rising long-term care costs, the Netherlands reformed its long-term care policy in 2015⁴.”

References

1. Wammes JD, Swait JD, de Bekker-Grob EW, et al. Dyadic Discrete Choice Experiments Enable Persons with Dementia and Informal Caregivers to Participate in Health Care Decision Making: A Mixed Methods Study. *Journal of Alzheimer's Disease* 2022(Preprint):1-10.
2. Wammes JD, Labrie NH, Agogo GO, et al. Persons with dementia and informal caregivers prioritizing care: A mixed-methods study. *Alzheimer's & Dementia: Translational Research & Clinical Interventions* 2021;7(1):e12193.

VERSION 2 – REVIEW

REVIEWER	Haroon, Muhammad The University of Queensland, Faculty of Medicine
REVIEW RETURNED	19-Nov-2023
GENERAL COMMENTS	This protocol reads well now. The authors have incorporated the suggested changes and they have addressed the concerns of the

	reviewers adequately. The revised version is much clearer, and this reviewer recommends it for publication. One thing which this reviewer suggests to the authors is that for the thematic analysis of the focus groups, the authors should look into the more recent research papers by Braun and Clarke. While their 2006 paper is still valid and provides the basis for reflexive thematic analysis, Braun and Clarke have written extensively since then about the method.
--	---

REVIEWER	WANG , Kailu The Chinese University of Hong Kong
REVIEW RETURNED	22-Nov-2023

GENERAL COMMENTS	The authors have well addressed my questions and concerns. I have no further questions or comments.
---

VERSION 2 – AUTHOR RESPONSE

Reviewer: 1

Dr. Muhammad Haroon, The University of Queensland

Comments to the Author:

This protocol reads well now. The authors have incorporated the suggested changes and they have addressed the concerns of the reviewers adequately. The revised version is much clearer, and this reviewer recommends it for publication.

Comment 1

One thing which this reviewer suggests to the authors is that for the thematic analysis of the focus groups, the authors should look into the more recent research papers by Braun and Clarke. While their 2006 paper is still valid and provides the basis for reflexive thematic analysis, Braun and Clarke have written extensively since then about the method.

Answer 1

Thank you for bringing the more recent work of Braun and Clarke to our attention. We have read their more recent articles about the Reflexive Thematic analysis. We will take this into consideration for the analysis of the focus groups and interviews with persons with dementia and informal caregivers with a migration background.

See methods, page 7

“Reflexive thematic analysis will be used to analyze and identify relevant patterns within the data (33, 34).”

Reviewer: 2

Dr. Kailu WANG , The Chinese University of Hong Kong

Comments to the Author:

The authors have well addressed my questions and concerns. I have no further questions or comments.

Reviewer: 1

Competing interests of Reviewer: None.

Reviewer: 2

Competing interests of Reviewer: None